# Uncovering the phenotypic consequences of multi-locus imprinting disturbances using genome-wide methylation analysis in genomic imprinting disorders

Hwa Young Kim[1], Choong Ho Shin[2], Chang Ho Shin[3], Jung Min Ko[4,5]*

1 Department of Pediatrics, Division of Pediatric Endocrinology and Metabolism, Seoul National University Bundang Hospital, Seongnam, Korea, 2 Department of Pediatrics, Division of Pediatric Endocrinology and Metabolism, Seoul National University College of Medicine, Seoul, Korea, 3 Department of Orthopaedics, Division of Pediatric Orthopedics, Seoul National University College of Medicine, Seoul, Korea, 4 Department of Pediatrics, Division of Clinical Genetics, Seoul National University College of Medicine, Seoul, Korea, 5 Rare Disease Center, Seoul National University Hospital, Seoul, Korea

* jmko@snu.ac.kr

**Data Availability Statement:** The entire data are available from the Gene Expression Omnibus (GEO) database (accession number GSE237503).

## Abstract

Imprinted genes are regulated by DNA methylation of imprinted differentially methylated regions (iDMRs). An increasing number of patients with congenital imprinting disorders (IDs) exhibit aberrant methylation at multiple imprinted loci, multi-locus imprinting disturbance (MLID). We examined MLID and its possible impact on clinical features in patients with IDs. Genome-wide DNA methylation analysis (GWMA) using blood leukocyte DNA was performed on 13 patients with Beckwith–Wiedemann syndrome (BWS), two patients with Silver–Russell syndrome (SRS), and four controls. HumanMethylation850 BeadChip analysis for 77 iDMRs (809 CpG sites) identified three patients with BWS and one patient with SRS showing additional hypomethylation, other than the disease-related iDMRs, suggestive of MLID. Two regions were aberrantly methylated in at least two patients with BWS showing MLID: *PPIEL* locus (chromosome 1: 39559298 to 39559744), and *FAM50B* locus (chromosome 6: 3849096 to 3849469). All patients with BWS- and SRS-MLID did not show any other clinical characteristics associated with additional involved iDMRs. Exome analysis in three patients with BWS who exhibited multiple hypomethylation did not identify any causative variant related to MLID. This study indicates that a genome-wide approach can unravel MLID in patients with an apparently isolated ID. Patients with MLID showed only clinical features related to the original IDs. Long-term follow-up studies in larger cohorts are warranted to evaluate any possible phenotypic consequences of other disturbed imprinted loci.

## Introduction

Genomic imprinting is an epigenetic regulatory mechanism, leading to a parent-of-origin-specific expression of a small subset of genes [1]. The life cycle of the genomic imprints in mammals consists of three stages: establishment of parental imprinting marks in the germline during gametogenesis, imprint maintenance through fertilization and early embryonic

**Funding:** JMK received the Seoul National University Hospital Research Funds (grant number 03-2022-0360) for this work. The funders had no role in study design, data collection and analysis, decision to publish, or preparation of the manuscript.

**Competing interests:** The authors have declared that no competing interests exist.

development, and erasure of imprints in primordial germ cells [2]. Dysregulation at one of these stages results in disturbances in expression of the imprinted genes, leading to imprinting disorders (IDs) [3]. The majority of the imprinted genes are found in clusters that contain CpG-rich regions with DNA methylation on the non-expressed allele, imprinted differentially methylated regions (iDMRs) [4]. Primary methylation defects at some well characterized iDMRs are responsible for 12 clinically characterized IDs in humans, majority of which show common clinical phenotypes of aberrant pre- and/or postnatal growth [5].

Imprinting disturbances in the chromosome 11p15.5 region, harboring two imprinted domains controlled by its own imprinting control regions (ICRs) (ICR1 for *H19/IGF2*: IG-DMR and ICR2 for *KCNQ1OT1*:TSS-DMR), are involved in two clinically opposing growth disorders: Beckwith–Wiedemann syndrome (BWS, OMIM #130650), an overgrowth disorder, and Silver–Russell syndrome (SRS, OMIM #180860), a growth retardation disorder [2]. Loss of methylation (LoM) at ICR1 (IC1-LoM) triggers SRS in 30%–60% of cases, while LoM at ICR2 (IC2-LoM) and gain of methylation (GoM) at ICR1 (IC1-GoM) accounts for 50% and 5% of cases with BWS, respectively. In addition, paternal uniparental disomy (UPD) of chromosome 11 and maternal UPD for chromosome 7 are present in 20% of BWS and 5%–10% of SRS, respectively [6, 7].

Current diagnostic techniques in IDs routinely assess disturbances at single, disease-associated regions based on distinct clinical findings. However, a growing number of patients with 11p15-related growth disorders have been reported to exhibit multi-locus imprinting disturbance (MLID), that is, abnormal methylation in other imprinting domains besides the disease-specific locus [8, 9]. Observations of MLID suggested the involvement of *trans*-acting factors in regulating imprinting marks involved in the establishment or post-fertilization imprint maintenance [10]. While several causative genes have been identified in a few patients with MLID, the underlying genetic etiology remains unknown in the majority of patients with MLID [11–15].

The phenotypic consequences of MLID remains controversial. Although most previously reported patients with MLID present only with clinical features of the original IDs, several studies reported peculiar features in patients with BWS and SRS showing MLID [9, 16]. According to recent data from 11 laboratories, the detection rates of MLID using commercial methylation-specific multiplex ligation-dependent probe amplification (MS-MLPA) kits were 12.7% in BWS with IC2-LoM, and 5.1% in SRS with IC1-LoM, but multi-locus tests were not conducted in all patients [17]. Currently, genome-wide methylation arrays (GWMA) have become possible, enabling exhaustively detect MLID in patients with ID [18–20]. In the present study, we investigated the presence of MLID and its possible clinical impact in patients with BWS and SRS using GWMA.

## Materials and methods

### Ethics approval

This study was approved by the Institutional Review Board of the Seoul National University Hospital (IRB Number: 2106-120-1230). All participants and their parents provided written informed consent prior to the study enrollment.

### Subjects

Thirteen patients with BWS and two patients with SRS were included. Epimutations at the chromosome 11p15.5 imprinting region were confirmed in all patients using MS-MLPA and bisulfite pyrosequencing analysis using leukocyte genomic DNA as previously described [21]. MS-MLPA was conducted using a SALSA MLPA kit (ME030 BWS/SRS, MRS Holland,

Amsterdam, Netherlands) following the manufacturer's instructions. Targeted bisulfite pyro-sequencing assays covering four and seven consecutive CpG sites for ICR1 and ICR2, respectively, were performed using a PyroMark Q24 pyrosequencer (Qiagen, Hilden, Germany) (S1 Table). Methylation levels were calculated as a percentage of methylated cytosine [% mC = mC/(mC+C)] for each CpG site using PyroMark Q24 Software (v.1.0.10; Qiagen). The standard deviation scores (SDS) of altered DNA methylation level (%mC) at ICR1 or ICR2 in patients with BWS and SRS was calculated based on the average DNA methylation level detected in 20 samples from age- and sex-matched controls.

For patients with BWS, the clinical score was calculated using the BWS consensus scoring system: 2 points per each cardinal features (macroglossia, exomphalos, lateralized overgrowth, multifocal Wilms tumor, prolonged hyperinsulinism, distinct pathologic findings), and 1 point per each suggestive features (large for gestational age, facial naevus flammeus, polyhydramnios or placentomegaly, ear creases or pits, transient hypoglycemia, embryonal tumors, nephromegaly or hepatomegaly and umbilical hernias or diastasis recti) [6]. Patients with score of ≥4 were regarded as classical BWS. For patients with SRS, the clinical score was calculated using the Netchine-Harbison clinical scoring system (NH–CSS), including small for gestational age, postnatal growth failure, relative macrocephaly at birth, protruding forehead, body asymmetry, feeding difficulties and/or low body mass index [7]. Age- and sex- specific SDS for serial growth parameters including height, weight, head circumference, and/or body mass index were assigned based on Fenton growth references at birth [22] and 2017 Korean National Growth Charts at postnatally [23]. Overgrowth or undergrowth were defined as height or weight greater than or less than two standard deviations compared to age- and sex-matched controls [6, 7]. Developmental and pubertal status was evaluated by a clinical geneticist and an endocrinologist, respectively. Skeletal abnormalities, including leg length discrepancy, scoliosis, and thoracic deformity, were assessed by an orthopedic surgeon using x-ray images. Clinical data for other phenotypes, including cardiac, gastrointestinal, and central nervous system anomalies, were retrospectively collected from medical records.

## GWMA and data analysis

A peripheral blood sample (5 mL) was collected in each participant at initial visit, and genomic DNA was extracted from all types of peripheral blood leukocytes including granular and agranular leukocytes using the DNA isolation kit (QIAGEN, Hilden, Germany). GWMA was performed in 13 patients with BWS, two patients with SRS, and four healthy children controls. Sodium bisulfite conversion of genomic DNA was conducted using an EZ DNA Methylation Kit (Zymo Research). After bisulfite conversion, each sample was whole-genome amplified, enzymatically fragmented, and applied to the Illumina HumanMethylation850 BeadChip (Illumina, Inc., San Diego, CA, USA), which contains probes at more than 850,000 CpG loci. The amplified DNA was annealed to allele-specific primers linked to two individual bead types, which correspond to each CpG locus—one methylated (C) and one unmethylated (T). After single-base extension using DNP- and biotin-labeled ddNTPs, the array was fluorescently labeled, followed by washing and coating. The Illumina BeadArray Reader, and Illumina GenomeStudio v. 2011.1 were used for image reading and extracting image intensities. The location of the CpG relative to UCSC CGI was assigned to each probe; shores (sequences of 0–2 kb from the CGI), shelves (sequences 2–4 kb from the CGI), and open sea (sequences located outside these regions) [24]. The CpGs were functionally classified as proximal promoters (CpGs located within 200 or 1500 bp upstream of transcription start sites, exon 1, and 5′ UTRs), 3′ UTRs, gene bodies, and intergenic regions [24]. Arrays were processed at Macrogen (Seoul, Korea) by following the manufacturer's instructions.

The $\beta$ values were extracted for each CpG by subtracting background intensity computed from negative controls and taking the ratio of the methylated signal intensity to the combined intensity of methylated and unmethylated signals ranging from 0 (no methylation) to 1 (100% methylation). From 865,918 CpGs in total, 865,544 CpGs were detected by detection $P$-value of <0.05. As a quality control step, we eliminated probes associated with single-nucleotide polymorphism, cross-hybridization probes, and probes that corresponded to the X or Y chromosome from the dataset, leaving 745,915 CpGs to be analyzed. The Beta Mixture Quantile (BMIQ) normalization of filtered data was conducted to reduce assay bias [25]. To confirm the abnormal methylation levels of disease-specific iDMRs and detect MLID in each patient, the Crawford–Howell $t$-test was performed using preprocessed data [26]. For 809 CpGs on 77 iDMRs defined by Monk [27] and Joshi [28], the DNA methylation level difference (delta beta, $\Delta\beta$) was calculated by subtracting mean $\beta$ value of the control group from $\beta$ value of each patient. Differentially methylated position (DMP) was defined as the absolute value of $\Delta\beta$ ($|\Delta\beta|$) of >0.2 (>20% difference in DNA methylation) and $P$-value of <0.05. Aberrantly methylated iDMRs were defined as iDMRs showing differentially methylated levels in at least two consecutive probes within iDMRs which included at least four significant probes [19]. The *VTRNA2–1*:DMR was considered as insignificant because it has been shown to be polymorphically imprinted in general population, showing frequent hypomethylation with various degrees [29]. We considered each patient as having MLID if showing aberrant methylation in one or more additional iDMRs other than the disease-related iDMRs.

## Exome sequencing

Three samples from patients with BWS exhibiting MLID were subjected to whole-exome sequencing (Fig 1). The exome sequencing data of the patient were analyzed using SureSelectXT Human All Exon V5 (Agilent Technologies, CA, USA), and the library was prepared according to the manufacturer's protocol. An Illumina HiSeq2500 platform was used for library sequencing with $2 \times 151$ paired-end reads. Reads were aligned to the human genome

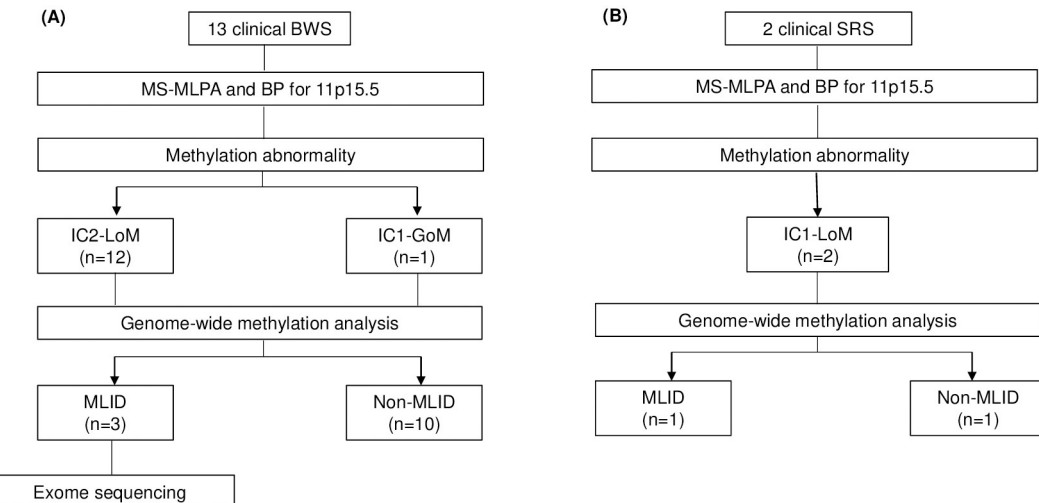

**Fig 1. Flowchart of the molecular tests.** Molecular tests for patients with BWS (A) and patients with SRS (B). BWS, Beckwith–Wiedemann syndrome; MS-MLPA, methylation-specific multiplex ligation-dependent probe amplification; BP, bisulfite pyrosequencing; IC2, imprinting center 2; LoM, loss of methylation; IC1, imprinting center 1; GoM, gain of methylation; MLID, multi-locus imprinting disturbance; SRS, Silver–Russell syndrome.

build 37 (hg19) using Burrows–Wheeler Aligner (v. 0.7.17), and PCR duplicates were filtered out using Picard software (v. 2.9.0). The binary alignment map (BAM) file was realigned and recalibrated using SAMtools (v. 1.9), and the Genome Analysis Toolkit (v. 4.1.2). Variants were called using HaplotypeCaller and annotated using SnpEff, ANNOVAR, and InterVar. Sequence variants were compared with databases such as Genome Aggregation Database (gnomAD), Human Gene Mutation Database, and ClinVar. For further analysis, variants with zero allele frequency in the gnomAD were selected for autosomal dominant genes, and variants with allele frequency <0.01% were selected for autosomal recessive genes. The low-frequency variants (allele frequency 0.05–0.25) were detected using MuTect2. Copy number variation and low-frequency variants were analyzed as described previously [30]. After variant classification using guidelines of the American College of Medical Genetics and Genomics guidelines, we considered pathogenic or likely pathogenic variants as causative variants [31]. Variant screening for candidate genes was performed and genes involved in imprinting establishment and maintenance during embryonic development were analyzed (S2 Table) [32, 33].

## Statistical analysis

Descriptive data are presented as mean ± standard deviation, and categorical variables as counts and proportions. The group differences according to the MLID (mono-locus vs. multi-locus) were investigated using the Mann–Whitney U test and Fisher's exact test. Statistical analysis was performed using SPSS v. 25.0 (SPSS Inc. Chicago, IL, USA) and R software v. 3.6.0 (The Comprehensive R Archive Network, Vienna, Austria; https://cran.r-project.org). A *P*-value of <0.05 was considered as statistically significant.

## Results

### Subjects characteristics

Table 1 shows the clinical and molecular characteristics of 13 patients with BWS (six males, seven females), and two patients with SRS (one male, one female). The mean follow-up duration was 4.6 ± 3.2 years in BWS, and 9.0 ± 2.3 years in SRS. Patients with BWS comprised two different epigenotypes (12 IC2-LoM, one IC1-GoM) and all met the criteria for classical BWS (a score of ≥4). The mean SDS for height and weight at initial visit (at a mean age of 1.8 ± 2.4 years) in patients with BWS were +2.2 ± 1.6, and +1.9 ± 1.1, respectively. Two patients with SRS had the IC1-LoM epigenotype and both met more than four out of six NH–CSS features. The SDS for height and weight at initial visit were –4.2, and –3.4 in SRS1 (at 3.6 months of age), and –4.1, and –7.2 in SRS2 (at 4.1 years of age), respectively. Both patients with SRS were treated with growth hormone, starting at a mean age of 4.3 years at a dosage for small for gestational age (0.3 mg/kg/week). SRS2 received combination therapy with a gonadotropin-releasing hormone agonist due to advanced puberty.

### Genome-wide alteration of DNA methylation

Hierarchical cluster analysis of the methylated region using GWMA identified 74 significant DMPs in 15 patients with ID compared with four healthy children controls: 29 (39.2%) hyper DMPs and 45 (60.8%) hypo DMPs. Patients with ID were generally classified into two categories based on the genome-wide methylation pattern: (1) category 1 of 12 patients with BWS (all with IC2-LoM); (2) category 2 of two patients with SRS (both with IC1-LoM) and one patient with BWS (with IC1-GoM) (Fig 2). In a single-sample analysis for each patient, the number of significantly hypermethylated DMPs per patient ranged from 175 (BWS8) to

**Table 1. Clinical and molecular characteristics of patients.**

| Patient | Age (years) | Sex | At initial visit | | | Clinical characteristics | | | Epigenotype | BP (11p15.5) | |
|---|---|---|---|---|---|---|---|---|---|---|---|
| | | | Age (years) | Ht (SDS) | Wt (SDS) | Disease-specific | Other | CS[a] | | IC1 (%mC SDS) | IC2 (%mC SDS) |
| BWS1 | 4.8 | F | 0.1 | 2.1 | 1.9 | S, L, MO, E, G, U | - | 8 | IC2-LoM | N | L (−12.1) |
| BWS2 | 8.8 | F | 7.6 | 0.5 | 0 | S, L, E, FN | MMD | 6 | IC2-LoM | N | L (−11.8) |
| BWS3 | 1.2 | F | 0.0 | 1.7 | 1.1 | S, E, O | ASD, PDA, BC | 5 | IC2-LoM | N | L (−18.1) |
| BWS4 | 3.2 | F | 0.1 | 4.4 | 2.7 | S, E, R | - | 5 | IC2-LoM | N | L (−19.1) |
| BWS5 | 4.8 | M | 1.5 | 1.4 | 2.9 | S, MO, E, U, FN | - | 5 | IC2-LoM | N | L (−19.6) |
| BWS6 | 5.9 | M | 0.7 | 5 | 2.9 | S, L, MO, E, U, T | - | 7 | IC2-LoM | N | L (−15.2) |
| BWS7 | 1.2 | M | 0.5 | 0.4 | 0.6 | S, MO, FN, O | I | 6 | IC2-LoM | N | L (−18.3) |
| BWS8 | 12.6 | F | 4.9 | 1.2 | 1.1 | S, E, O | ASD, CH | 5 | IC2-LoM | N | L (−19.3) |
| BWS9 | 9.3 | F | 0.1 | 2.7 | 3.4 | S, L, MO, O, T | - | 8 | IC2-LoM | N | L (−18.8) |
| BWS10 | 5.5 | F | 2.0 | 0.9 | 1.6 | S, L, E, U, FN | - | 7 | IC2-LoM | N | L (−15.0) |
| BWS11 | 8.1 | M | 4.7 | 0.6 | 0.7 | S, L, U, FN, P | - | 8 | IC2-LoM | N | L (−19.0) |
| BWS12 | 10.1 | M | 0.5 | 4.5 | 3.4 | S, L, E, U, FN | I | 7 | IC2-LoM | N | L (−16.9) |
| BWS13 | 8.8 | M | 0.1 | 3.6 | 1.9 | S, MO, E, G, U, T, P | VSD, AP | 8 | IC1-GoM | H (+6.3) | N |
| SRS1 | 7.7 | F | 0.3 | −4.3 | −3.4 | A, GF, RM, L | C, SUA, PT | 4 | IC1-LoM | L (−2.9) | N |
| SRS2 | 14.7 | M | 4.1 | −4.1 | −7.2 | A, GF, RM, L, LBMI | C, TF, LE | 5 | IC1-LoM | L (−3.1) | N |

[a]Consensus scoring system for BWS [6]; Netchine-Harbison clinical scoring system for SRS [7].

Abbreviations SDS, standard deviation score; Ht, height; Wt, weight; CS, clinical score; BP, Bisulfite pyrosequencing; IC1, imprinting center 1; %mC, percentage of methylated cytosine; IC2, imprinting center 2; MLID, multi-locus methylation disturbance; BWS, Beckwith-Wiedemann syndrome; F, female; S, macroglossia; L, body asymmetry; MO, macrosomia; E, ear anomalies; G, organomegaly; N, normal; L, low; U, umbilical hernia or diastasis recti; LoM, loss of methylation; FN, facial nevus simplex; MMD, moyamoya disease; O, exomphalos; ASD, atrial septal defect; PDA, patent ductus arteriosus; BC, branchial cleft cyst; R, nephrogenic rest; M, male; T, transient hypoglycemia; CH, congenital hypothyroidism; I, Inguinal hernia; P, polyhydramnios or placentomegaly; VSD, ventricular septal defect; AP, advanced puberty; GoM, gain of methylation; H, high; SRS, Silver-Russell syndrome; A, small for gestational age; GF, postnatal growth failure; RM, relative macrocephaly; C, clinodactyly on the 5th finger; SUA, single umbilical artery; PT, premature thelarche; LBMI, low body mass index; TF, triangular face; LE, low-set ears.

17,035 (BWS7), while the number of significantly hypomethylated DMPs ranged from 375 (SRS2) to 3,760 (BWS7) (Table 2).

When focusing on 809 CpGs in 77 iDMRs defined by Monk et al. [27] and Joshi et al. [28], a total of 232 CpGs in 17 iDMRs (two paternally and 15 maternally methylated) were identified as differentially methylated in patients with ID, compared with the control group (mean methylation level of 0.4–0.6). All patients with BWS and SRS showed aberrant methylation in disease-related iDMRs, which were consistent with those identified in targeting analysis using bisulfite pyrosequencing (Table 1). In addition, the GWMA data revealed 14 additional hypomethylation other than the disease-related iDMRs, suggesting MLID in four patients: three patients with BWS (BWS8, BWS9, and BWS12), and one patient with SRS (SRS1) (S3 Table).

## Multi-locus methylation defects

Three patients with BWS-MLID (BWS8, BWS9, and BWS12) had two or more additional LoM-type epimutations at the maternally methylated iDMRs. In BWS9, MLID was the most extensive, with LoM at seven additional iDMRs (*PPIEL*:Ex1-DMR, *DIRAS3*:Ex2-DMR, *NAP1L5*:TSS-DMR, *FAM50B*:TSS-DMR, *PLAGL1*:alt-TSS-DMR, *IGF1R*:Int2-DMR, and *WRB*:alt-TSS-DMR) (Fig 3). A total of 102 CpGs in 12 additional iDMRs were hypomethylated in three patients with BWS-MLID; 52.9% in CpG islands (CGIs), 25.5% in CpG shores, and 21.6% in outside these regions (open sea) (S4 Table). Among those, 14 CpGs in two hypomethylated iDMRs were aberrantly methylated in at least two patients with BWS-MLID;

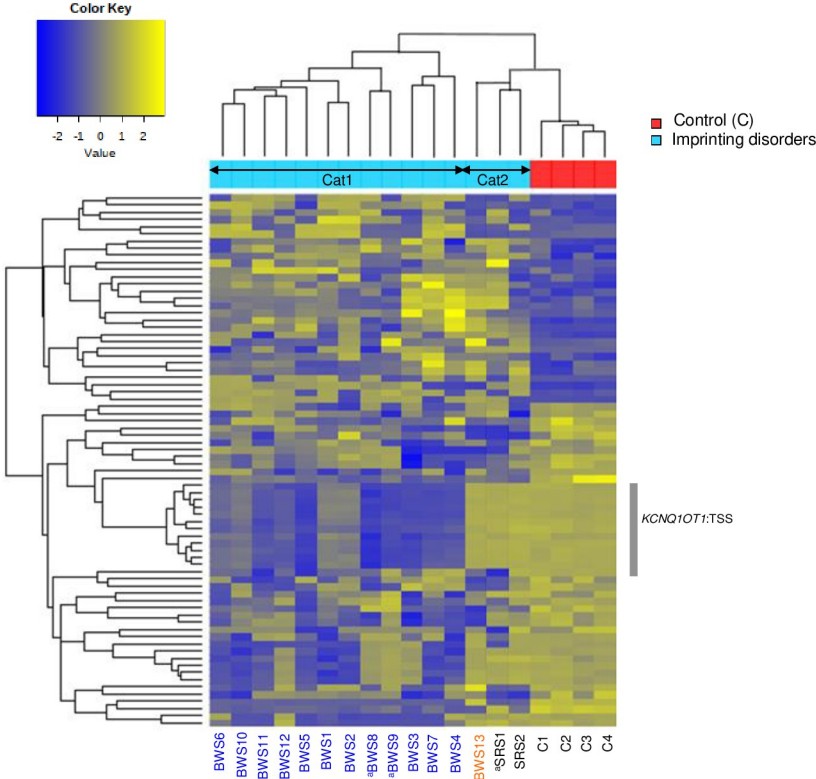

**Fig 2. Hierarchical clustering of differentially methylated CpG sites in 15 ID patients versus four controls.**
Methylation values are extracted from the 850K array. Clustering of 74 significant CpGs satisfying with the average
DNA methylation level difference between groups >0.2 with a P value <0.05. For BWS patients, two epigenotypes are
indicated by color (IC2-LoM, blue; IC1-GoM, orange). BWS, Beckwith-Wiedemann syndrome; SRS, Silver-Russell
syndrome; C, control; IC2, imprinting center 2; LoM, loss of methylation; IC1, imprinting center 1; GoM, gain of
methylation. [a]Multi-locus methylation disturbance.

*PPIEL* locus (chromosome 1: 39559298 to 39559744) with a methylation difference of 0.24 and
4 significant probes, and *FAM50B* locus (chromosome 6: 3849096 to 3849469) with a methyla-
tion difference of 0.26 and 10 significant probes. Relative to CpG context, 71.4% of the sites
were in CGIs, and 28.6% were located in open sea (S5 Table). In SRS1 who showed MLID,
LoM was identified at 58 CpGs in two additional iDMRs: paternally- and maternally methyl-
ated iDMRs (*MEG3*:TSS-DMR and *PEG10*:TSS-DMR, respectively); 69.0% in CGIs and 31.0%
in CpG shores (S6 Table).

## Clinical features of patients with and without MLID

All patients with MLID showed the classical phenotype of BWS or SRS without any clinically
distinguishable features (Table 1). BWS9 who showed additional LoM of *PLAGL1*:alt-
TSS-DMR in 6q24.2, implicated in the etiology of transient neonatal diabetes mellitus-1
(OMIM #601410), did not exhibit hyperglycemia until 9.3 years of age. Likewise, there was not
any synergic effect for growth failure in SRS1 who had additional LoM of *MEG3*:TSS-DMR in
14q32.2, which is molecularly corresponding to Temple syndrome (OMIM #616222). Among
12 patients with BWS with IC2-LoM, there were no significant differences in clinical features
according to the presence of MLID, in terms of sex, mode of conception (assisted reproductive

**Table 2. Significant DMPs per patient in 850K.**

| Patient | Epigenotype | Number of hypermethylated DMPs | % hypermethylated DMPs in 850 K | Number of hypomethylated DMPs | % hypomethylated DMPs in 850 K |
|---|---|---|---|---|---|
| BWS1 | IC2-LoM | 3,171 | 0.425 | 523 | 0.070 |
| BWS2 | IC2-LoM | 374 | 0.050 | 630 | 0.084 |
| BWS3 | IC2-LoM | 2,870 | 0.385 | 3,134 | 0.420 |
| BWS4 | IC2-LoM | 14,061 | 1.885 | 1,050 | 0.141 |
| BWS5 | IC2-LoM | 261 | 0.035 | 590 | 0.079 |
| BWS6 | IC2-LoM | 1,702 | 0.228 | 813 | 0.109 |
| BWS7 | IC2-LoM | 17,035 | 2.284 | 3,760 | 0.504 |
| BWS8[a] | IC2-LoM | 175 | 0.023 | 435 | 0.058 |
| BWS9[a] | IC2-LoM | 302 | 0.040 | 577 | 0.077 |
| BWS10 | IC2-LoM | 688 | 0.092 | 846 | 0.113 |
| BWS11 | IC2-LoM | 2,642 | 0.354 | 802 | 0.108 |
| BWS12[a] | IC2-LoM | 1,041 | 0.140 | 596 | 0.080 |
| BWS13 | IC1-GoM | 1,370 | 0.184 | 1,344 | 0.180 |
| SRS1[a] | IC1-LoM | 857 | 0.115 | 801 | 0.107 |
| SRS2 | IC1-LoM | 162 | 0.022 | 375 | 0.050 |

[a]Multi-locus methylation disturbance.

Abbreviations DMPs, differentially methylated positions; BWS, Beckwith-Wiedemann syndrome; IC2, imprinting center 2; LoM, loss of methylation; IC1, imprinting center 1; GoM, gain of methylation; SRS, Silver-Russell syndrome.

technique), parental age at conception, prenatal and postnatal growth parameters, BWS-related phenotypes, and clinical score (Table 3).

## Exome analysis of putative MLID-causative variants

In order to investigate the putative MLID-causative variants, whole-exome sequencing using blood leukocyte DNA was performed in three patients with BWS who exhibited multiple

| CHR | iDMRs | Methylation | BWS1 IC2-LoM | BWS2 IC2-LoM | BWS3 IC2-LoM | BWS4 IC2-LoM | BWS5 IC2-LoM | BWS6 IC2-LoM | BWS7 IC2-LoM | BWS8 IC2-LoM | BWS9 IC2-LoM | BWS10 IC2-LoM | BWS11 IC2-LoM | BWS12 IC2-LoM | BWS13 IC1-GoM | SRS1 IC1-LoM | SRS2 IC1-LoM |
|---|---|---|---|---|---|---|---|---|---|---|---|---|---|---|---|---|---|
| 1 | PPIEL:Ex1 | Maternal | | | | | | | | | 0.23(4) | | | 0.25(4) | | | |
| | DIRAS3:TSS | Maternal | | | | | | | | | 0.25(8) | | | | | | |
| 4 | NAP1L5:TSS | Maternal | | | | | | | | | 0.37(13) | | | | | | |
| 6 | FAM50B:TSS | Maternal | | | | | | | | | 0.27(8) | | | 0.22(9) | | | |
| | PLAGL1:alt-TSS | Maternal | | | | | | | | | 0.38(14) | | | | | | |
| 7 | PEG10:TSS | Maternal | | | | | | | | | | | | | | 0.34(44) | |
| 11 | H19/IGF2:IG | Paternal | | | | | | | | | | | | | 0.33(35) | 0.26(16) | 0.29(30) |
| | KCNQ1OT1:TSS | Maternal | 0.27(10) | 0.25(16) | 0.38(23) | 0.35(23) | 0.41(23) | 0.31 (19) | 0.35(23) | 0.42(23) | 0.38(23) | 0.28(17) | 0.35(23) | 0.34(22) | | | |
| 14 | MEG3:TSS | Paternal | | | | | | | | | | | | | | 0.24(14) | |
| 15 | IGF1R:Int2 | Maternal | | | | | | | | | 0.36(5) | | | | | | |
| 19 | ZNF331:alt-TSS-1 | Maternal | | | | | | | | | | | | 0.22(5) | | | |
| | ZNF331:alt-TSS-2 | Maternal | | | | | | | | | | | | 0.25(7) | | | |
| 20 | L3MBTL1:alt-TSS | Maternal | | | | | | | | 0.26(16) | | | | | | | |
| | GNAS-AS1:TSS | Maternal | | | | | | | | 0.24(12) | | | | | | | |
| 21 | WRB:alt-TSS | Maternal | | | | | | | | | 0.28(4) | | | | | | |
| 22 | NHP2L1:alt-TSS | Maternal | | | | | | | | | | | | 0.27(4) | | | |

**Fig 3. Methylation changes at differentially methylated iDMRs in ID patients.** Orange and blue boxes indicate hypermethylated and hypomethylated iDMRs, respectively. Numbers in the boxes indicate |Δβ| (number of probes). CHR, chromosome; iDMRs, imprinting-associated differentially methylated regions; BWS, Beckwith-Wiedemann syndrome; SRS, Silver-Russell syndrome; IC2, imprinting center 2; LoM, loss of methylation; IC1, imprinting center 1; GoM, gain of methylation.

**Table 3. Comparisons of clinical features of BWS patients with IC2-LoM with and without MLID.**

| | Mono-locus (N = 9) | Multi-locus (N = 3) | *P*-value |
|---|---|---|---|
| Female, N (%) | 5 (55.6) | 2 (66.7) | 1.000 |
| Gestational age (weeks) | 36.6 ± 2.8 | 38.7 ± 1.5 | 0.255 |
| Assisted reproductive technique, N (%) | 1 (11.1) | 1 (33.3) | 0.455 |
| Paternal age at conception (year) | 37.3 ± 3.7 | 36.5 ± 0.7 | 0.772 |
| Maternal age at conception (year) | 35.0 ± 5.5 | 33.5 ± 0.7 | 0.724 |
| Birth length (SDS) | +1.8 ± 1.4 | +1.2 ± 0.4 | 0.619 |
| Birth weight (SDS) | +1.6 ± 0.6 | +1.7 ± 0.9 | 0.880 |
| Birth head circumference (SDS) | +0.6 ± 0.4 | +1.4 ± 0.5 | 0.080 |
| Height at last visit (SDS) | +1.6 ± 0.7 | +1.9 ± 0.5 | 0.548 |
| Weight at last visit (SDS) | +1.4 ± 0.9 | +1.4 ± 0.8 | 0.985 |
| Head circumference at last visit (SDS) | −0.0 ± 0.0 | +0.0 ± 0.0 | 0.098 |
| Clinical score | 6.3 ± 1.2 | 6.7 ± 1.5 | 0.707 |
| Macroglossia, N (%) | 9 (100) | 3 (100) | NA |
| Exomphalos, N (%) | 3 (33.3) | 2 (66.7) | 0.523 |
| Lateralized overgrowth, N (%) | 5 (55.6) | 2 (66.7) | 1.000 |
| Wilms tumor or nephroblastomatosis, N (%) | 1 (11.1) | 0 (0.0) | 1.000 |
| Macrosomia (birthweight >2 SDS), N (%) | 3 (33.3) | 1 (33.3) | 1.000 |
| Facial nevus simplex, N (%) | 5 (55.6) | 1 (33.3) | 1.000 |
| Polyhydramnios or placentomegaly, N (%) | 1 (11.1) | 0 (0.0) | 1.000 |
| Ear creases or pits, N (%) | 7 (77.8) | 2 (66.7) | 1.000 |
| Transient hypoglycemia, N (%) | 1 (11.1) | 1 (33.3) | 0.455 |
| Nephromegaly or hepatomegaly, N (%) | 1 (11.1) | 0 (0.0) | 1.000 |
| Umbilical hernia or diastasis recti, N (%) | 3 (33.3) | 1 (33.3) | 1.000 |

Mean ± SD, n(%).

Abbreviations BWS, Beckwith-Wiedemann syndrome; IC2, imprinting center 2; LoM, loss of methylation; MLID, multi-locus imprinting disturbance; SDS, standard deviation score.

hypomethylation. No copy number variant was found in the exome data of three patients. In addition, no causative variants were identified among candidate genes, including *KHDC3L*, *NLRP2*, *NLRP5*, *NLRP7*, *ZFP57* [32, 33].

## Discussion

Using GWMA, we found four patients with MLID (three BWS and one SRS). MLID was mostly revealed in LoM-type epimutations at the maternally methylated iDMRs, in accordance with previous studies [5, 8, 9, 34–36]. LoM predominance at the maternally methylated iDMRs could be explained by the fact that the majority of iDMRs are of maternal origin [37]. It is also possible that the maternally imprinted loci may be more susceptible to LoM than paternally imprinted loci in the establishment and maintenance of methylation during gametogenesis [34, 38]. However, concurrent hypomethylation of both paternally- and maternally methylated iDMRs (*MEG3* and *PEG10*, respectively) was observed in SRS1 in our study. Coexistence of paternal and maternal LoM was also reported in previous reports [8, 39, 40], which suggested the postzygotic origin of epigenetic defects, as opposed to occurrence in germ-line cells.

The involved iDMRs varied in our patients with MLID, supporting non-recurrent methylation defects in MLID [33]. All 12 additional iDMRs in our patients with BWS-MLID

were previously described in patients with BWS, SRS, and other IDs showing MLID [9, 19, 20, 33, 39, 41]. Likewise, two additional iDMRs (*MEG3* and *PEG10*) in our patient with SRS-MLID were also previously reported in patients with SRS showing MLID [16, 42]. Among three patients with BWS-MLID, we found two hypomethylated iDMRs (*PPIEL*: Ex1-DMR, and *FAM50B*:TSS-DMR) in at least two patients. Aberrant DNA methylation at *PPIEL* and *FAM50B* have been associated with bipolar disorder, and intellectual disability, respectively [18, 43]. However, the clinical consequence of these affected loci is uncertain because our patients with BWS-MLID did not exhibit these previously reported clinical features.

Our patients with BWS- and SRS-MLID did not show any other clinical features due to hypomethylation of other loci, implying an epi-dominant effect of one locus above the others in clinical phenotypes [8]. Likewise, most patients with MLID showed only clinical features related to the original IDs [8, 34, 44]. Nonetheless, several cases showed atypical phenotypes such as developmental delay and congenital anomalies [9, 13, 16, 33, 45]. This epigenotype–phenotype divergence in patients with MLID possibly reflects somatic mosaicism, that is, the degree of methylation disturbances at the critical CpGs within other iDMRs remained at a subclinical level in the target tissues [5]. However, the influence of additionally involved iDMRs cannot be completely excluded. As MLID-associated clinical signs may manifest as patients grow up, careful monitoring is warranted to determine the possible effect of additional disturbances. For example, patients with LoM at the *PLAGL1* locus require monitoring for early-onset diabetes. Besides, routine screening for childhood cancers may be required in patients with LoM at the *DIRAS3* iDMR, which is associated with imprinted tumor-associated genes [46].

In our study, no causative variants of MLID have been found in exome sequencing analysis in three probands with BWS who exhibited multiple LoM. Until now, variants affecting either zygotic or oocyte-derived *trans*-acting factors have rarely been reported in some patients with MLID or in their mothers [47]. In particular, recessive variants of *ZFP57*, which prevent demethylation of genomic iDMRs during early embryogenesis, have been identified in seven of 13 probands with TNDM showing MLID [13]. Maternal-effect variants affecting components of the subcortical maternal complex, which plays an important role in imprinting establishment, have also been reported in women with reproductive problems, such as hydatidiform mole and miscarriages, and in healthy women with offspring with MLID and IDs, including BWS [15, 48, 49]. Convincing evidence has been provided for the involvement of *NLRP2*, *NLRP5*, *NLRP7*, and *KHDC3L* in the etiology of IDs, while the role of other maternal-effect genes, including *PADI6*, *OOEP*, *TLE6*, *UHRF1*, and *ZAR1* is not definitely established [48]. A detailed investigation in a large number of patients with MLID may provide further insights into the imprint acquisition and post-fertilization maintenance of imprinted DNA methylation.

This study has some limitations. First, we determined the methylation status in only blood leukocytes, and possible effect of tissue mosaicism could not be excluded. Second, our study focused only on the DNA methylation patterns analyzed in the 850K array. Methylation defects at other non-investigated regions may also involve the complex genome-wide methylation phenomenon. Third, due to the limited number of patients identified with MLID, and relatively short follow-up duration, the functional consequences of disruptions at other imprinted loci are still unclear. Nevertheless, this study was strengthened by the extensive analysis of the epigenome in deeply phenotyped patients with ID with proven epimutation. We have expanded the spectrum of epimutated iDMRs associated with MLID in patients with BWS and SRS.

## Conclusions

Using GWMA, we identified MLID in three patients with BWS and one patient with SRS. Additional hypomethylated iDMRs varied in the number and degree of affected regions in individual patients. Patients with MLID did not show any clinically distinct characteristics. However, the long-term phenotypic impact of other disturbed imprinted loci remains to be elucidated via expansion of patient cohort.

## Supporting information

**S1 Table. Primers used for bisulfite pyrosequencing.**
(XLSX)

**S2 Table. Candidate genes included in variant screening.**
(XLSX)

**S3 Table. Significant differentially methylated regions in each patient with IDs, compared with the controls.**
(XLSB)

**S4 Table. Additional differentially methylated CpG sites identified in BWS-MLID patients, compared with the controls.**
(XLSX)

**S5 Table. Methylation changes in shared affected iDMRs in BWS-MLID patients.**
(XLSX)

**S6 Table. Additional differentially methylated CpG sites identified in an SRS-MLID patient, compared with the controls.**
(XLSX)

## Acknowledgments

We express our gratitude to the patients and their parents for their participation in the study.

## Author Contributions

**Conceptualization:** Hwa Young Kim, Choong Ho Shin, Chang Ho Shin, Jung Min Ko.

**Data curation:** Hwa Young Kim, Chang Ho Shin, Jung Min Ko.

**Formal analysis:** Hwa Young Kim.

**Funding acquisition:** Jung Min Ko.

**Investigation:** Hwa Young Kim, Chang Ho Shin, Jung Min Ko.

**Methodology:** Hwa Young Kim, Choong Ho Shin, Chang Ho Shin, Jung Min Ko.

**Resources:** Choong Ho Shin, Jung Min Ko.

**Supervision:** Choong Ho Shin, Jung Min Ko.

**Writing – original draft:** Hwa Young Kim.

**Writing – review & editing:** Choong Ho Shin, Chang Ho Shin, Jung Min Ko.

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
