## [Decision Letter · Decision Letter 0]

11 Apr 2023

PONE-D-23-04262Uncovering the phenotypic consequences of multi-locus imprinting disturbances using genome-wide methylation analysis in genomic imprinting disordersPLOS ONE

Dear Dr. Ko,

Thank you for submitting your manuscript to PLOS ONE. After careful consideration, we feel that it has merit but does not fully meet PLOS ONE’s publication criteria as it currently stands. Therefore, we invite you to submit a revised version of the manuscript that addresses the points raised during the review process.

We look forward to receiving your revised manuscript.

Kind regards,

Osman El-Maarri, Ph.D

Academic Editor

PLOS ONE

Reviewers' comments:

Reviewer's Responses to Questions

**Comments to the Author**

1. Is the manuscript technically sound, and do the data support the conclusions?

Reviewer #1: Yes

Reviewer #2: No

2. Has the statistical analysis been performed appropriately and rigorously? 

Reviewer #1: Yes

Reviewer #2: No

3. Have the authors made all data underlying the findings in their manuscript fully available?

Reviewer #1: Yes

Reviewer #2: No

4. Is the manuscript presented in an intelligible fashion and written in standard English?

Reviewer #1: Yes

Reviewer #2: Yes

5. Review Comments to the Author

Reviewer #1: Ku et al. conducted array-based methylation analysis in 13 patients with BWS caused by LOM of IC2 or GOM of IC1 and two patients with SRS caused by LOM of IC1, and examined their genome-wide alteration of DNA methylation and multi-locus methylation defects and compared clinical features between patients with MLID and without MLID. Their paper is important for understanding clinical features and methylation abnormalities of MLID. However, the reviewer has some comments.

Comment 1

Did the authors conduct mutation screening for the known MLID causative genes in the patients with MLID? Even if the negative data, these results are very informative.

Comments 2

The reviewer wants to know the clinical features of patients with SRS. The patient with MLID has hypomethylation of the H19-DMR and MEG3-DMR. The reviewer also wants to know whether there is a synergic effect for growth failure or not in this patient.

Comment 3

Recently, D Mackay et al. published a paper about MLID (2022 Clinical Epigenetics). This study included patients with MLID detected in multiple centers in the world. Please cite this paper for the frequency of MLID and related descriptions.

Comment 4

In table 1, the authors described %mC SDS. Because IC1 and IC2 include many CpGs, the authors need to describe the calculation method of %mC SDS.

Comment 5

Please describe the kind of MS-MLPA kit used in this study.

Comment 6

Line 101-102, 182

Regarding GWMA controls, the authors mentioned "four age- and sex-matched control samples ", but it is unclear whether you used four samples per each case or four samples overall. If it is four samples overall, most of these cases are not age and sex-matched. Please describe these matters in detail.

Comment 7

Line 276-279

The authors described that VTRNA2-1 is associated with cancer risk in adulthood. Because of non-adult cases in this study, the description of "our patients with BWS-MLID did not exhibit these previously reported clinical features." is not appropriate.

Comment 8

Line 284-287

Please cite a suitable reference.

Comment 9

Line 290-291

The authors need to additionally describe VTRNA2-1 here.

Comment 10

Line 104

“x1000” is not clear. Is this means “ whole genome amplification”? The authors need to modify this sentence.

Reviewer #2: The authors claimed to have identified five out of 13 BWS cases and one out of two SRS cases as MLID cases by examining their methylation patterns at known imprinted DMRs using Illumina’s EPIC array.

Major comments:

1. The data the authors presented in the Figures and Tables including supplementary ones are limited to a subset of imprinted DMR probes. At least, the authors should provide beta values for all of 809 CG probes for 77 iDMRs for all BWS/SRS/control individuals. Otherwise, it is difficult for readers to precisely compare this study with previous relevant studies.

The authors have included beta value data of only hypomethylated probes in BWS and SRS patients and shown the average of delta beta values of those probes in Figure 3. To more precisely show the extent of hypomethylation of each of DMRs including ICR1 and ICR2, beta values of all probes within such hypomethylated DMRs should be plotted.

The authors should also deposit the entire and individual 850K-array data to a public database. If “some restrictions will apply” as the authors declared for the Data Availability, the authors should consider depositing their data to a control-access database such as the database of Genotypes and Phenotypes (dbGAP) and European Genome-Phenome Archive (EGA).

2. The VTRNA2-1/nc886 locus has been shown to be polymorphically imprinted in the populations (Marttila S et al. Epigenomics. 2022 PMID: 36200237). See Figure 4A (blood, n=2664) of Marttila S et al. (PMID: 36200237). In BWS10 and BWS11 cases, VTRNA2-1 was the only hypomethylated DMR other than KCNQ1OT1:TSS-DMR. Because hypomethylation at VTRNA2-1-DMR is frequently observed in the general population with various degrees ranging from complete loss of methylation to partial hypomethylation, this reviewer thinks that the authors should not classify BWS10 and BWS11 as MLID cases. The authors need to mention to the polymorphic nature of VTRNA2-1 imprinting reported by many previous papers in the Introduction or the Discussion section. If the authors classify BWS10 and BWS11 as MLID cases, they need to describe the rationale for it.

6. PLOS authors have the option to publish the peer review history of their article (what does this mean?). If published, this will include your full peer review and any attached files.

Reviewer #1: No

Reviewer #2: No

---

## [Author Response · Author response to Decision Letter 0]

18 Jul 2023

> We have checked the revised manuscript to meet PLOS ONE's style requirements.

> We have deposited the entire data and related metadata collected in our study in a public data repository for microarray data, Gene Expression Omnibus (GEO), and included the relevant GEO accession number (GSE237503) in our revised cover letter as well as in the Data Availability Statement. The data will remain private until publication and a 'private access token' for distribution to journal reviewers has been created as follows: sxmlcwqgdlevvwl.

> Thank you for your comment. Full ethics statement, including the full name of the IRB and the information that written consent was obtained from the participants, was described in the Material and Methods section (page 4, line 82-85). 

Reviewers' comments:

Reviewer's Responses to Questions

Comments to the Author

1. Is the manuscript technically sound, and do the data support the conclusions?

Reviewer #1: Yes

Reviewer #2: No

2. Has the statistical analysis been performed appropriately and rigorously?

Reviewer #1: Yes

Reviewer #2: No

3. Have the authors made all data underlying the findings in their manuscript fully available?

Reviewer #1: Yes

Reviewer #2: No

4. Is the manuscript presented in an intelligible fashion and written in standard English?

Reviewer #1: Yes

Reviewer #2: Yes

5. Review Comments to the Author

Reviewer #1: Ku et al. conducted array-based methylation analysis in 13 patients with BWS caused by LOM of IC2 or GOM of IC1 and two patients with SRS caused by LOM of IC1, and examined their genome-wide alteration of DNA methylation and multi-locus methylation defects and compared clinical features between patients with MLID and without MLID. Their paper is important for understanding clinical features and methylation abnormalities of MLID. However, the reviewer has some comments.

Comment 1

Did the authors conduct mutation screening for the known MLID causative genes in the patients with MLID? Even if the negative data, these results are very informative.

> Thank you for your constructive comment. We have conducted exome analysis to investigate the causative variants related to MLID in three patients with BWS who exhibited multiple hypomethylation. However, no causative variants were identified among candidate genes, including KHDC3L, NLRP2, NLRP5, NLRP7, ZFP57. 

We have added the descriptions about our molecular investigation for the putative MLID-causative variants in the abstract (page 2, line 32-34), the materials and methods section (page 8, line 163; page 9, line 183), and the result section (page 15, line 304; page 16, line 309), with additional descriptions of previously reported MLID causative genes in the introduction (page 4, line 65-69), and the discussion section (page 17, line 348-362).

Comments 2

The reviewer wants to know the clinical features of patients with SRS. The patient with MLID has hypomethylation of the H19-DMR and MEG3-DMR. The reviewer also wants to know whether there is a synergic effect for growth failure or not in this patient.

> Thank you for your comment. Aggravation of growth failure was not observed in SRS1 who showed additional hypomethylation of MEG3:TSS-DMR (molecularly corresponding to Temple syndrome). We have additionally described this point in the results section (page 14, line 290-293).

Comment 3

Recently, D Mackay et al. published a paper about MLID (2022 Clinical Epigenetics). This study included patients with MLID detected in multiple centers in the world. Please cite this paper for the frequency of MLID and related descriptions.

> Thank you for your helpful comment. We have cited the paper that the reviewer mentioned in the introduction section (page 4, line 73-76), with the MLID detection rates and related descriptions presented in that paper.

Comment 4

In table 1, the authors described %mC SDS. Because IC1 and IC2 include many CpGs, the authors need to describe the calculation method of %mC SDS.

> Thank you for your comment. We have described the calculation method of %mC SDS at IC1 and IC2 in our internally designed targeted assays covering four and seven consecutive CpG sites for each locus in the materials and methods section (page 5, line 92-99), and presented information about primers used for bisulfite pyrosequencing analysis as Supplementary Table 1.

Comment 5

Please describe the kind of MS-MLPA kit used in this study.

> Thank you for your comment. We have described the kind of MS-MLPA kit (ME030 BWS/SRS) used in our study in the materials and methods section (page 5, line 91-92).

Comment 6

Line 101-102, 182

Regarding GWMA controls, the authors mentioned "four age- and sex-matched control samples ", but it is unclear whether you used four samples per each case or four samples overall. If it is four samples overall, most of these cases are not age and sex-matched. Please describe these matters in detail.

> Thank you for your comment. We used four healthy control samples overall (two males, two females). We used leukocyte genomic DNA collected at the patients’ initial visit for GWMA analysis. Accordingly, control samples matching age at blood collection according to sex in the patient group (in terms of mean and standard deviation) were used for comparison. However, we agree with the reviewer’s comment that it is difficult to mention that most of cases were age- and sex-matched because the ranges of ages at blood collection were wide in our patient group. We have corrected the time for blood collection in the materials and methods section (page 6, line 122) and changed the description of control group for GWMA as ‘healthy children controls’, instead of ‘age- and sex-matched healthy controls’ in the materials and methods section (page 6, line 125-126) and the results section (page 11, line 229).

Comment 7

Line 276-279

The authors described that VTRNA2-1 is associated with cancer risk in adulthood. Because of non-adult cases in this study, the description of "our patients with BWS-MLID did not exhibit these previously reported clinical features." is not appropriate.

> Thank you for your comment. According to the reviewer#2’s comments that the VTRNA2-1/nc886 locus is polymorphic in general population, two BWS cases with only one additional hypomethylation at the VTRNA2-1:DMR (BWS10 and BWS11) were reclassified as non-MLID cases in the revised manuscript. Accordingly, we have removed previous description of the phenotypic consequences of VTRNA2-1 locus in the discussion section.

Comment 8

Line 284-287

Please cite a suitable reference.

> Thank you for your comment. We have cited a relevant reference.

Comment 9

Line 290-291

The authors need to additionally describe VTRNA2-1 here.

> Thank you for your comment. As mentioned in our response to the reviewer’s comment 7, we considered the hypomethylation at the VTRNA2-1:DMR to be insignificant due to it’s polymorphic nature of imprinting which was demonstrated in a recent study (Epigenomics 2022: PMID 36200237). We have added relevant information about the VTRNA2-1:DMR along with reference#29 in the methods section (page 8, line 157-159). 

Comment 10

Line 104

“x1000” is not clear. Is this means “ whole genome amplification”? The authors need to modify this sentence.

> Thank you for your comment. For clear meaning, the expression ‘x1000’ was modified to ‘whole-genome amplified’.

Reviewer #2: The authors claimed to have identified five out of 13 BWS cases and one out of two SRS cases as MLID cases by examining their methylation patterns at known imprinted DMRs using Illumina’s EPIC array.

Major comments:

1. The data the authors presented in the Figures and Tables including supplementary ones are limited to a subset of imprinted DMR probes. At least, the authors should provide beta values for all of 809 CG probes for 77 iDMRs for all BWS/SRS/control individuals. Otherwise, it is difficult for readers to precisely compare this study with previous relevant studies.

The authors have included beta value data of only hypomethylated probes in BWS and SRS patients and shown the average of delta beta values of those probes in Figure 3. To more precisely show the extent of hypomethylation of each of DMRs including ICR1 and ICR2, beta values of all probes within such hypomethylated DMRs should be plotted.

The authors should also deposit the entire and individual 850K-array data to a public database. If “some restrictions will apply” as the authors declared for the Data Availability, the authors should consider depositing their data to a control-access database such as the database of Genotypes and Phenotypes (dbGAP) and European Genome-Phenome Archive (EGA).

> Thank you for your comment. In addition to Figure 3 which schematically displays aberrantly methylated iDMRs, we have additionally presented beta values for all significant differentially methylated positions for each BWS/SRS/control individuals in supplementary Table (S3 Table). We also have deposited the entire data and related metadata collected in our study in a public data repository for microarray data, Gene Expression Omnibus (accession number GSE237503), and a 'private access token' for distribution to journal reviewers until publication has been created as follows: sxmlcwqgdlevvwl.

2. The VTRNA2-1/nc886 locus has been shown to be polymorphically imprinted in the populations (Marttila S et al. Epigenomics. 2022 PMID: 36200237). See Figure 4A (blood, n=2664) of Marttila S et al. (PMID: 36200237). In BWS10 and BWS11 cases, VTRNA2-1 was the only hypomethylated DMR other than KCNQ1OT1:TSS-DMR. Because hypomethylation at VTRNA2-1-DMR is frequently observed in the general population with various degrees ranging from complete loss of methylation to partial hypomethylation, this reviewer thinks that the authors should not classify BWS10 and BWS11 as MLID cases. The authors need to mention to the polymorphic nature of VTRNA2-1 imprinting reported by many previous papers in the Introduction or the Discussion section. If the authors classify BWS10 and BWS11 as MLID cases, they need to describe the rationale for it.

> Thank you for your constructive comment. We agree with the reviewer’s opinion that the VTRNA2-1/nc886 locus should not be considered as meaningful iDMR in imprinting disorders because the imprinting mark in this locus is polymorphic in general population. 

We have mentioned that the VTRNA2-1:DMR was considered as insignificant, with a description of it’s polymorphic nature in the methods section (page 8, line 157-159). 

In addition, two BWS cases with only one additional hypomethylation at the VTRNA2-1:DMR (BWS10 and BWS11) were excluded from MLID cases. Accordingly, we have changed the descriptions for the number of patients with BWS showing MLID and related results throughout the revised manuscript, including Table 2, Table 3, Fig. 1, Fig. 2, Fig. 3, S4 Table, and S5 Table.

6. PLOS authors have the option to publish the peer review history of their article (what does this mean?). If published, this will include your full peer review and any attached files.

Do you want your identity to be public for this peer review? For information about this choice, including consent withdrawal, please see our Privacy Policy.

Reviewer #1: No

Reviewer #2: No

---

## [Decision Letter · Decision Letter 1]

9 Aug 2023

Uncovering the phenotypic consequences of multi-locus imprinting disturbances using genome-wide methylation analysis in genomic imprinting disorders

PONE-D-23-04262R1

Dear Dr. Ko,

We’re pleased to inform you that your manuscript has been judged scientifically suitable for publication and will be formally accepted for publication once it meets all outstanding technical requirements.

Kind regards,

Osman El-Maarri, Ph.D

Academic Editor

PLOS ONE

Additional Editor Comments (optional):

Reviewers' comments:

Reviewer's Responses to Questions

**Comments to the Author**

1. If the authors have adequately addressed your comments raised in a previous round of review and you feel that this manuscript is now acceptable for publication, you may indicate that here to bypass the “Comments to the Author” section, enter your conflict of interest statement in the “Confidential to Editor” section, and submit your "Accept" recommendation.

Reviewer #1: All comments have been addressed

Reviewer #2: All comments have been addressed

2. Is the manuscript technically sound, and do the data support the conclusions?

Reviewer #1: Yes

Reviewer #2: Yes

3. Has the statistical analysis been performed appropriately and rigorously? 

Reviewer #1: Yes

Reviewer #2: Yes

4. Have the authors made all data underlying the findings in their manuscript fully available?

Reviewer #1: Yes

Reviewer #2: Yes

5. Is the manuscript presented in an intelligible fashion and written in standard English?

Reviewer #1: No

Reviewer #2: Yes

6. Review Comments to the Author

Reviewer #1: (No Response)

Reviewer #2: The authors addressed my concerns regarding data availability and the interpretation for the methylation of vtRNA2-1 DMR appropriately.

7. PLOS authors have the option to publish the peer review history of their article (what does this mean?). If published, this will include your full peer review and any attached files.

Reviewer #1: No

Reviewer #2: No

---

## [Editor Report · Acceptance letter]

11 Aug 2023

PONE-D-23-04262R1 

Uncovering the phenotypic consequences of multi-locus imprinting disturbances using genome-wide methylation analysis in genomic imprinting disorders 

Dear Dr. Ko:

I'm pleased to inform you that your manuscript has been deemed suitable for publication in PLOS ONE. Congratulations! Your manuscript is now with our production department. 

Kind regards, 

on behalf of

Priv.-Doz. Dr. Osman El-Maarri 

Academic Editor

PLOS ONE